# DEEP CONTINUOUS CLUSTERING

## ABSTRACT

Clustering high-dimensional datasets is hard because interpoint distances become less informative in high-dimensional spaces. We present a clustering algorithm that performs nonlinear dimensionality reduction and clustering jointly. The data is embedded into a lower-dimensional space by a deep autoencoder. The autoencoder is optimized as part of the clustering process. The resulting network produces clustered data. The presented approach does not rely on prior knowledge of the number of ground-truth clusters. Joint nonlinear dimensionality reduction and clustering are formulated as optimization of a global continuous objective. We thus avoid discrete reconfigurations of the objective that characterize prior clustering algorithms. Experiments on datasets from multiple domains demonstrate that the presented algorithm outperforms state-of-the-art clustering schemes, including recent methods that use deep networks.

## 1 INTRODUCTION

Clustering is a fundamental procedure in machine learning and data analysis. Well-known approaches include center-based methods and their generalizations (Banerjee et al., 2005; Teboulle, 2007), and spectral methods (Ng et al., 2001; von Luxburg, 2007). Despite decades of progress, reliable clustering of noisy high-dimensional datasets remains an open problem. High dimensionality poses a particular challenge because assumptions made by many algorithms break down in high-dimensional spaces (Ball, 1997; Beyer et al., 1999; Steinbach et al., 2004).

There are techniques that reduce the dimensionality of data by embedding it in a lower-dimensional space (van der Maaten et al., 2009). Such general techniques, based on preserving variance or dissimilarity, may not be optimal when the goal is to discover cluster structure. Dedicated algorithms have been developed that combine dimensionality reduction and clustering by fitting low-dimensional subspaces (Kriegel et al., 2009; Vidal, 2011). Such algorithms can achieve better results than pipelines that first apply generic dimensionality reduction and then cluster in the reduced space. However, frameworks such as subspace clustering and projected clustering operate on linear subspaces and are therefore limited in their ability to handle datasets that lie on nonlinear manifolds.

Recent approaches have sought to overcome this limitation by constructing a nonlinear embedding of the data into a low-dimensional space in which it is clustered (Dizaji et al., 2017; Xie et al., 2016; Yang et al., 2016; 2017). Ultimately, the goal is to perform nonlinear embedding and clustering jointly, such that the embedding is optimized to bring out the latent cluster structure. These works have achieved impressive results. Nevertheless, they are based on classic center-based, divergence-based, or hierarchical clustering formulations and thus inherit some limitations from these classic methods. In particular, these algorithms require setting the number of clusters a priori. And the optimization procedures they employ involve discrete reconfigurations of the objective, such as discrete reassignments of datapoints to centroids or merging of putative clusters in an agglomerative procedure. Thus it is challenging to integrate them with an optimization procedure that modifies the embedding of the data itself.

We seek a procedure for joint nonlinear embedding and clustering that overcomes some of the limitations of prior formulations. There are a number of characteristics we consider desirable. First, we wish to express the joint problem as optimization of a single continuous objective. Second, this optimization should be amenable to scalable gradient-based solvers such as modern variants of SGD. Third, the formulation should not require setting the number of clusters a priori, since this number is often not known in advance.

While any one of these desiderata can be fulfilled by some existing approaches, the combination is challenging. For example, it has long been known that the $k$-means objective can be optimized by SGD (Bottou & Bengio, 1994). But this family of formulations requires positing the number of clusters $k$ in advance. Furthermore, the optimization is punctuated by discrete reassignments of datapoints to centroids, and is thus hard to integrate with continuous embedding of the data.

In this paper, we present a formulation for joint nonlinear embedding and clustering that possesses all of the aforementioned desirable characteristics. Our approach is rooted in Robust Continuous Clustering (RCC), a recent formulation of clustering as continuous optimization of a robust objective (Shah & Koltun, 2017). The basic RCC formulation has the characteristics we seek, such as a clear continuous objective and no prior knowledge of the number of clusters. However, integrating it with deep nonlinear embedding is still a challenge. For example, Shah & Koltun (2017) presented a formulation for joint *linear* embedding and clustering (RCC-DR), but this formulation relies on a complex alternating optimization scheme with linear least-squares subproblems, and does not apply to nonlinear embeddings.

We present an integration of the RCC objective with dimensionality reduction that is simpler and more direct than RCC-DR, while naturally handling deep nonlinear embeddings. Our formulation avoids alternating optimization and the introduction of auxiliary dual variables. A deep nonlinear embedding of the data into a low-dimensional space is optimized while the data is clustered in the reduced space. The optimization is expressed by a global continuous objective and conducted by standard gradient-based solvers.

The presented algorithm is evaluated on high-dimensional datasets of images and documents. Experiments demonstrate that our formulation performs on par or better than state-of-the-art clustering algorithms across all datasets. This includes recent approaches that utilize deep networks and rely on prior knowledge of the number of ground-truth clusters. Controlled experiments confirm that joint dimensionality reduction and clustering is more effective than a stagewise approach, and that the high accuracy achieved by the presented algorithm is stable across different dimensionalities of the latent space.

## 2 PRELIMINARIES

Let $\mathbf{X} = [\mathbf{x}_1, \ldots, \mathbf{x}_N]$ be a set of points in $\mathbb{R}^D$ that must be clustered. Generic clustering algorithms that operate directly on $\mathbf{X}$ rely strongly on interpoint distances. When $D$ is high, these distances become less informative (Ball, 1997; Beyer et al., 1999). Hence most clustering algorithms do not operate effectively in high-dimensional spaces. To overcome this problem, we embed the data into a lower-dimensional space $\mathbb{R}^d$. The embedding of the dataset into $\mathbb{R}^d$ is denoted by $\mathbf{Y} = [\mathbf{y}_1, \ldots, \mathbf{y}_N]$. The function that performs the embedding is denoted by $f_{\boldsymbol{\theta}} : \mathbb{R}^D \to \mathbb{R}^d$. Thus $\mathbf{y}_i = f_{\boldsymbol{\theta}}(\mathbf{x}_i)$ for all $i$.

Our goal is to cluster the embedded dataset $\mathbf{Y}$ and to optimize the parameters $\boldsymbol{\theta}$ of the embedding as part of the clustering process. This formulation presents an obvious difficulty: if the embedding $f_{\boldsymbol{\theta}}$ can be manipulated to assist the clustering of the embedded dataset $\mathbf{Y}$, there is nothing that prevents $f_{\boldsymbol{\theta}}$ from distorting the dataset such that $\mathbf{Y}$ no longer respects the structure of the original data. We must therefore introduce a regularizer on $\boldsymbol{\theta}$ that constrains the low-dimensional image $\mathbf{Y}$ with respect to the original high-dimensional dataset $\mathbf{X}$. To this end, we also consider a reverse mapping $g_{\boldsymbol{\omega}} : \mathbb{R}^d \to \mathbb{R}^D$. To constrain $f_{\boldsymbol{\theta}}$ to construct a faithful embedding of the original data, we require that the original data be reproducible from its low-dimensional image (Hinton & Salakhutdinov, 2006):

$$\underset{\boldsymbol{\Omega}}{\text{minimize}} \ \|\mathbf{X} - G_{\boldsymbol{\omega}}(\mathbf{Y})\|_F^2, \qquad \text{where } \mathbf{Y} = F_{\boldsymbol{\theta}}(\mathbf{X}), \quad \boldsymbol{\Omega} = \{\boldsymbol{\theta}, \boldsymbol{\omega}\}. \tag{1}$$

Here $F_{\boldsymbol{\theta}}(\mathbf{X}) = [f_{\boldsymbol{\theta}}(\mathbf{x}_1), \ldots, f_{\boldsymbol{\theta}}(\mathbf{x}_N)]$, $G_{\boldsymbol{\omega}}(\mathbf{Y}) = [g_{\boldsymbol{\omega}}(\mathbf{y}_1), \ldots, g_{\boldsymbol{\omega}}(\mathbf{y}_N)]$, and $\|\cdot\|_F$ denotes the Frobenius norm.

Next, we must decide how the low-dimensional embedding $\mathbf{Y}$ will be clustered. A natural solution is to choose a classic clustering framework: a center-based method such as $k$-means, a divergence-based formulation, or an agglomerative approach. These are the paths taken in recent work on combining nonlinear dimensionality reduction and clustering (Dizaji et al., 2017; Xie et al., 2016; Yang et al., 2016; 2017). However, the classic clustering algorithms have a discrete structure: associations between centroids and datapoints need to be recomputed or putative clusters need to be merged. In either case, the optimization process is punctuated by discrete reconfigurations. This makes it difficult

to coordinate the clustering of $\mathbf{Y}$ with the optimization of the embedding parameters $\boldsymbol{\Omega}$ that modify the dataset $\mathbf{Y}$ itself.

Since we must conduct clustering in tandem with continuous optimization of the embedding, we seek a clustering algorithm that is inherently continuous and performs clustering by optimizing a continuous objective that does not need to be updated during the optimization. The recent RCC formulation provides a suitable starting point (Shah & Koltun, 2017). The key idea of RCC is to introduce a set of representatives $\mathbf{Z} \in \mathbb{R}^{d \times N}$ and optimize the following nonconvex objective:

$$\underset{\mathbf{Z}}{\text{minimize}} \ \frac{1}{2}\|\mathbf{Z} - \mathbf{Y}\|_F^2 + \frac{\lambda}{2} \sum_{(i,j) \in \mathcal{E}} w_{i,j}\rho(\|\mathbf{z}_i - \mathbf{z}_j\|_2), \tag{2}$$

where $\rho$ is a redescending M-estimator, $\mathcal{E}$ is a graph connecting the datapoints, $\{w_{i,j}\}$ are appropriately defined weights, and $\lambda$ is a coefficient that balances the two objective terms. The first term in objective (2) constrains the representatives to remain near the corresponding datapoints. The second term pulls the representatives to each other, encouraging them to merge. This formulation has a number of advantages. First, it reduces clustering to optimization of a fixed continuous objective. Second, each datapoint has its own representative in $\mathbf{Z}$ and no prior knowledge of the number of clusters is needed. Third, the nonconvex robust estimator $\rho$ limits the influence of outliers.

To perform nonlinear embedding and clustering jointly, we wish to integrate the reconstruction objective (1) and the RCC objective (2). This idea is developed in the next section.

## 3 DEEP CONTINUOUS CLUSTERING

### 3.1 OBJECTIVE

The Deep Continuous Clustering (DCC) algorithm optimizes the following objective:

$$\mathcal{L}(\boldsymbol{\Omega}, \mathbf{Z}) = \frac{1}{D} \underbrace{\|\mathbf{X} - G_{\boldsymbol{\omega}}(\mathbf{Y})\|_F^2}_{\text{reconstruction loss}} + \frac{1}{d}\left( \underbrace{\sum_i \rho_1\big(\|\mathbf{z}_i - \mathbf{y}_i\|_2; \mu_1\big)}_{\text{data loss}} + \lambda \underbrace{\sum_{(i,j) \in \mathcal{E}} w_{i,j}\rho_2\big(\|\mathbf{z}_i - \mathbf{z}_j\|_2; \mu_2\big)}_{\text{pairwise loss}} \right)$$

where $\mathbf{Y} = F_{\boldsymbol{\theta}}(\mathbf{X})$. $\tag{3}$

This formulation bears some similarity to RCC-DR (Shah & Koltun, 2017), but differs in three major respects. First, RCC-DR only operates on a linear embedding defined by a sparse dictionary, while DCC optimizes a more expressive nonlinear embedding parameterized by $\boldsymbol{\Omega}$. Second, RCC-DR alternates between optimizing dictionary atoms, sparse codes, representatives $\mathbf{Z}$, and dual line process variables; in contrast, DCC avoids duality altogether and optimizes the global objective directly. Third, DCC does not rely on closed-form or linear least-squares solutions to subproblems; rather, the joint objective is optimized by modern gradient-based solvers, which are commonly used for deep representation learning and are highly scalable.

We now discuss objective (3) and its optimization in more detail. The mappings $F_{\boldsymbol{\theta}}$ and $G_{\boldsymbol{\omega}}$ are performed by an autoencoder with fully-connected or convolutional layers and rectified linear units after each affine projection (Hinton & Salakhutdinov, 2006; Nair & Hinton, 2010). The graph $\mathcal{E}$ is constructed on $\mathbf{X}$ using the mutual kNN criterion (Brito et al., 1997), augmented by the minimum spanning tree of the kNN graph to ensure connectivity to all datapoints. The role of M-estimators $\rho_1$ and $\rho_2$ is to pull the representatives of a true underlying cluster into a single point, while disregarding spurious connections across clusters. For both estimators, we use scaled Geman-McClure functions (Geman & McClure, 1987):

$$\rho_1(x; \mu_1) = \frac{\mu_1 x^2}{\mu_1 + x^2} \quad \text{and} \quad \rho_2(x; \mu_2) = \frac{\mu_2 x^2}{\mu_2 + x^2}. \tag{4}$$

The parameters $\mu_1$ and $\mu_2$ control the radii of the convex basins of the estimators. The weights $w_{i,j}$ are set to balance the contribution of each datapoint to the pairwise loss:

$$w_{i,j} = \frac{\frac{1}{N}\sum_{k=1}^n n_k}{\sqrt{n_i n_j}}. \tag{5}$$

Here $n_i$ is the degree of $\mathbf{z}_i$ in the graph $\mathcal{E}$. The numerator is simply the average degree. The parameter $\lambda$ balances the relative strength of the data loss and the pairwise loss. To balance the different terms, we set $\lambda = \frac{\|\mathbf{Y}\|_2}{\|\mathbf{A}\|_2}$, where $\mathbf{A} = \sum_{(i,j)\in\mathcal{E}} w_{i,j}(\mathbf{e}_i - \mathbf{e}_j)(\mathbf{e}_i - \mathbf{e}_j)^\top$ and $\|\cdot\|_2$ denotes the spectral norm. In contrast to RCC-DR, the parameter $\lambda$ need not be updated during the optimization.

## 3.2 OPTIMIZATION

Objective (3) can be optimized using scalable modern forms of stochastic gradient descent (SGD). Note that each $\mathbf{z}_i$ is updated only via its corresponding loss and pairwise terms. On the other hand, the autoencoder parameters $\mathbf{\Omega}$ are updated via all data samples. Thus in a single epoch, there is bound to be a difference between the update rates for $\mathbf{Z}$ and $\mathbf{\Omega}$. To deal with this imbalance, an adaptive solver such as Adam should be used (Kingma & Ba, 2015).

Another difficulty is that the graph $\mathcal{E}$ connects all datapoints such that a randomly sampled minibatch is likely to be connected by pairwise terms to datapoints outside the minibatch. In other words, the objective (3), and more specifically the pairwise loss, does not trivially decompose over datapoints. This requires some care in the construction of minibatches. Instead of sampling datapoints, we sample subsets of edges from $\mathcal{E}$. The corresponding minibatch $\mathcal{B}$ is defined by all nodes incident to the sampled edges. However, if we simply restrict the objective (3) to the minibatch and take a gradient step, the reconstruction and data terms will be given additional weight since the same datapoint can participate in different minibatches, once for each incident edge. To maintain balance between the terms, we must weigh the contribution of each datapoint in the minibatch. The rebalanced minibatch loss is given by

$$\mathcal{L}_\mathcal{B}(\mathbf{\Omega}, \mathbf{Z}) = \frac{1}{|\mathcal{B}|} \sum_{i\in\mathcal{B}} w_i \left( \frac{\|\mathbf{x}_i - g_{\boldsymbol{\omega}}(\mathbf{y}_i)\|_2^2}{D} + \frac{\rho_1(\|\mathbf{z}_i - \mathbf{y}_i\|_2)}{d} \right) + \frac{\lambda}{|\mathcal{B}|} \sum_{(i,j)\in\mathcal{E}_\mathcal{B}} w_{i,j} \rho_2(\|\mathbf{z}_i - \mathbf{z}_j\|_2)$$

$$\text{where } \mathbf{y}_i = f_{\boldsymbol{\theta}}(\mathbf{x}_i) \quad \forall i \in \mathcal{B}. \tag{6}$$

Here $w_i = \frac{n_i^\mathcal{B}}{n_i}$, where $n_i^\mathcal{B}$ is the number of edges connected to the $i^{\text{th}}$ node in the subgraph $\mathcal{E}_\mathcal{B}$.

The gradients of $\mathcal{L}_\mathcal{B}$ with respect to the low-dimensional embedding $\mathbf{Y}$ and the representatives $\mathbf{Z}$ are given by

$$\frac{\partial \mathcal{L}_\mathcal{B}}{\partial \mathbf{y}_i} = \frac{1}{|\mathcal{B}|} \left( \frac{w_i \mu_1^2 (\mathbf{y}_i - \mathbf{z}_i)}{d(\mu_1 + \|\mathbf{z}_i - \mathbf{y}_i\|_2^2)^2} + \frac{2w_i(g_{\boldsymbol{\omega}}(\mathbf{y}_i) - \mathbf{x}_i)}{D} \frac{\partial g_{\boldsymbol{\omega}}(\mathbf{y}_i)}{\partial \mathbf{y}_i} \right) \tag{7}$$

$$\frac{\partial \mathcal{L}_\mathcal{B}}{\partial \mathbf{z}_i} = \frac{1}{|\mathcal{B}|} \left( \frac{w_i \mu_1^2 (\mathbf{z}_i - \mathbf{y}_i)}{d(\mu_1 + \|\mathbf{z}_i - \mathbf{y}_i\|_2^2)^2} + \lambda \mu_2^2 \sum_{(i,j)\in\mathcal{E}_\mathcal{B}} \frac{w_{i,j}(\mathbf{z}_i - \mathbf{z}_j)}{(\mu_2 + \|\mathbf{z}_i - \mathbf{z}_j\|_2^2)^2} \right) \tag{8}$$

These gradients are propagated to the parameters $\mathbf{\Omega}$.

## 3.3 INITIALIZATION, CONTINUATION AND STOPPING CRITERION

**Initialization.** The embedding parameters $\mathbf{\Omega}$ are initialized using the stacked denoising autoencoder (SDAE) framework (Vincent et al., 2010). Each pair of corresponding encoding and decoding layers is pretrained in turn. Noise is introduced during pretraining by adding dropout to the input of each affine projection (Srivastava et al., 2014). Encoder-decoder layer pairs are pretrained sequentially, from the outer to the inner. After all layer pairs are pretrained, the entire SDAE is fine-tuned end-to-end using the reconstruction loss. This completes the initialization of the embedding parameters $\mathbf{\Omega}$. These parameters are used to initialize the representatives $\mathbf{Z}$, which are set to $\mathbf{Z} = \mathbf{Y} = F_{\boldsymbol{\theta}}(\mathbf{X})$.

**Continuation.** The price of robustness is the nonconvexity of the estimators $\rho_1$ and $\rho_2$. One way to alleviate the dangers of nonconvexity is to use a continuation scheme that gradually sharpens the estimator (Blake & Zisserman, 1987; Mobahi & Fisher III, 2015). Following Shah & Koltun (2017), we initially set $\mu_i$ to a high value that makes the estimator $\rho_i$ effectively convex in the relevant range. The value of $\mu_i$ is decreased on a regular schedule until a threshold $\frac{\delta_i}{2}$ is reached. We set $\delta_1$ to the mean of the distance of each $\mathbf{y}_i$ to the mean of $\mathbf{Y}$, and $\delta_2$ to the mean of the bottom $1\%$ of the pairwise distances in $\mathcal{E}$ at initialization.

---

**Algorithm 1** Deep Continuous Clustering

1: **input**: Data samples $\{\mathbf{x}_i\}_i$.
2: **output**: Cluster assignment $\{c_i\}_i$.
3: Construct a graph $\mathcal{E}$ on $\mathbf{X}$.
4: Initialize $\mathbf{\Omega}$ and $\mathbf{Z}$.
5: Precompute $\lambda, w_{i,j}, \delta_1, \delta_2$. Initialize $\mu_1, \mu_2$.
6: **while** *stopping criterion not met* **do**
7:     Every iteration, construct a minibatch $\mathcal{B}$ defined by a sample of edges $\mathcal{E}_{\mathcal{B}}$.
8:     Update $\{\mathbf{z}_i\}_{i \in \mathcal{B}}$ and $\mathbf{\Omega}$.
9:     Every $M$ epochs, update $\mu_i = \max\left(\frac{\mu_i}{2}, \frac{\delta_i}{2}\right)$.
10: **end while**
11: Construct graph $\mathcal{G} = (\mathcal{V}, \mathcal{F})$ with $f_{i,j} = 1$ if $\|\mathbf{z}_i^* - \mathbf{z}_j^*\|_2 < \delta_2$.
12: Output clusters given by the connected components of $\mathcal{G}$.

---

**Stopping criterion.** Once the continuation scheme is completed, DCC monitors the computed clustering. At the end of every epoch, a graph $\mathcal{G} = (\mathcal{V}, \mathcal{F})$ is constructed such that $f_{i,j} = 1$ if $\|\mathbf{z}_i - \mathbf{z}_j\| < \delta_2$. The cluster assignment is given by the connected components of $\mathcal{G}$. DCC compares this cluster assignment to the one produced at the end of the preceding epoch. If less than $0.1\%$ of the edges in $\mathcal{E}$ changed from intercluster to intracluster or vice versa, DCC outputs the computed clustering and terminates.

**Complete algorithm.** The complete algorithm is summarized in Algorithm 1.

## 4 EXPERIMENTS

### 4.1 DATASETS

We conduct experiments on six high-dimensional datasets, which cover domains such as handwritten digits, objects, faces, and text. We used datasets from Shah & Koltun (2017) that had dimensionality above 100. The datasets are further described in the appendix. All features are normalized to the range $[0, 1]$.

Note that DCC is an unsupervised learning algorithm. Unlabelled data is embedded and clustered with no supervision. There is thus no train/test split.

### 4.2 BASELINES

The presented DCC algorithm is compared to 12 baselines, which include both classic and deep clustering algorithms. The baselines include $k$-means++ (Arthur & Vassilvitskii, 2007), DBSCAN (Ester et al., 1996), two variants of agglomerative clustering: Ward (AC-W) and graph degree linkage (GDL) (Zhang et al., 2012), two variants of spectral clustering: spectral embedded clustering (SEC) (Nie et al., 2011) and local discriminant models and global integration (LDMGI) (Yang et al., 2010), and two variant of robust continuous clustering: RCC and RCC-DR (Shah & Koltun, 2017).

The deep clustering baselines include four recent approaches that share our basic motivation and use deep networks for clustering: deep embedded clustering (DEC) (Xie et al., 2016), joint unsupervised learning (JULE) (Yang et al., 2016), the deep clustering network (DCN) (Yang et al., 2017), and deep embedded regularized clustering (DEPICT) (Dizaji et al., 2017). These are strong baselines that use deep autoencoders, the same network structure as our approach (DCC). The key difference is in the loss function and the consequent optimization procedure. The prior formulations are built on KL-divergence clustering, agglomerative clustering, and $k$-means, which involve discrete reconfiguration of the objective during the optimization and rely on knowledge of the number of ground-truth clusters either in the design of network architecture, during the embedding optimization, or in post-processing. In contrast, DCC optimizes a robust continuous loss and does not rely on prior knowledge of the number of clusters.

### 4.3 Implementation

We report experimental results for two different autoencoder architectures: one with only fully-connected layers and one with convolutional layers. This is motivated by prior deep clustering algorithms, some of which used fully-connected architectures and some convolutional.

For fully-connected autoencoders, we use the same autoencoder architecture as DEC (Xie et al., 2016). Specifically, for all experiments on all datasets, we use an autoencoder with the following dimensions: D–500–500–2000–d–2000–500–500–D. This autoencoder architecture follows parametric t-SNE (van der Maaten, 2009).

For convolutional autoencoders, the network architecture is modeled on JULE (Yang et al., 2016). The architecture is specified in the appendix. As in Yang et al. (2016), the number of layers depends on image resolution in the dataset and it is set such that the output resolution of the encoder is about $4 \times 4$.

In both architectures and for all datasets, the dimensionality of the reduced space is set to $d = 10$. (It is only varied for controlled experiments that analyze stability with respect to $d$.) No dataset-specific hyperparameter tuning was done. For autoencoder initialization, a minibatch size of 256 and dropout probability of 0.2 are used. SDAE pretraining and finetuning start with a learning rate of 0.1, which is decreased by a factor of 10 every 80 epochs. Each layer is pretrained for 200 epochs. Finetuning of the whole SDAE is performed for 400 epochs. For the fully-connected SDAE, the learning rates are scaled in accordance with the dimensionality of the dataset.

For m-kNN graph construction, the nearest-neighbor parameter $k$ is set to 10 and the cosine distance metric is used. The Adam solver is used with its default learning rate of 0.001 and momentum 0.99. Minibatches are constructed by sampling 128 edges. DCC was implemented using the PyTorch library.

For the baselines, we use publicly available implementations. For $k$-means++, DBSCAN and AC-W, we use the implementations in the SciPy library and report the best results across ten random restarts. For a number of baselines, we performed hyperparameter search to maximize their reported performance. For DBSCAN, we searched over values of $Eps$, for LDMGI we searched over values of the regularization constant $\lambda$, for SEC we searched over values of the parameter $\mu$, and for GDL we tuned the graph construction parameter $a$.

The DCN approach uses a different network architecture for each dataset. Wherever possible, we report results using their dataset-specific architecture. For YTF, Coil100, and YaleB, we use their reference architecture for MNIST.

### 4.4 Measures

Common measures of clustering accuracy include normalized mutual information (NMI) (Strehl & Ghosh, 2002) and clustering accuracy (ACC). However, NMI is known to be biased in favor of fine-grained partitions and ACC is also biased on imbalanced datasets (Vinh et al., 2010). To overcome these biases, we use adjusted mutual information (AMI) (Vinh et al., 2010), defined as

$$\text{AMI}(\mathbf{c}, \hat{\mathbf{c}}) = \frac{\text{MI}(\mathbf{c}, \hat{\mathbf{c}}) - E[\text{MI}(\mathbf{c}, \hat{\mathbf{c}})]}{\sqrt{\text{H}(\mathbf{c})\text{H}(\hat{\mathbf{c}})} - E[\text{MI}(\mathbf{c}, \hat{\mathbf{c}})]}. \tag{9}$$

Here $\text{H}(\cdot)$ is the entropy, $\text{MI}(\cdot, \cdot)$ is the mutual information, and $\mathbf{c}$ and $\hat{\mathbf{c}}$ are the two partitions being compared. AMI lies in a range $[0, 1]$. Higher is better. For completeness, results according to ACC are reported in the appendix.

### 4.5 Results

The results are summarized in Table 1. Among deep clustering methods that use fully-connected networks, DCN and DEC are not as accurate as fully-connected DCC and are also less consistent: the performance of DEC drops on the high-dimensional image datasets, while DCN is far behind on MNIST and YaleB. Among deep clustering methods that use convolutional networks, the performance of DEPICT drops on COIL100 and YTF, while JULE is far behind on YTF. The GDL algorithm failed to scale to the full MNIST dataset and the corresponding measurement is marked as 'n/a'.

| Algorithm | MNIST | Coil100 | YTF | YaleB | Reuters | RCV1 |
|---|---|---|---|---|---|---|
| $k$-means++ | 0.500 | 0.803 | 0.783 | 0.615 | 0.516 | 0.355 |
| AC-W | 0.679 | 0.853 | 0.801 | 0.767 | 0.471 | 0.364 |
| DBSCAN | 0.000 | 0.399 | 0.739 | 0.456 | 0.011 | 0.014 |
| SEC | 0.469 | 0.849 | 0.745 | 0.849 | 0.498 | 0.069 |
| LDMGI | 0.761 | 0.888 | 0.518 | 0.945 | 0.523 | 0.382 |
| GDL | n/a | 0.958 | 0.655 | 0.924 | 0.401 | 0.020 |
| RCC | 0.893 | 0.957 | 0.836 | 0.975 | 0.556 | 0.138 |
| RCC-DR | 0.828 | 0.957 | 0.874 | 0.974 | 0.553 | 0.442 |
| Fully-connected | | | | | | |
| DCN | 0.570 | 0.810 | 0.790 | 0.590 | 0.430 | 0.470 |
| DEC | 0.840 | 0.611 | 0.807 | 0.000 | 0.397 | **0.500** |
| DCC | **0.912** | 0.952 | 0.877 | 0.955 | **0.572** | 0.495 |
| Convolutional | | | | | | |
| JULE | 0.900 | **0.979** | 0.574 | **0.990** | – | – |
| DEPICT | **0.919** | 0.667 | 0.785 | **0.989** | – | – |
| DCC | **0.913** | 0.962 | **0.903** | **0.985** | – | – |

Table 1: Clustering accuracy of DCC and 12 baselines, measured by AMI. Higher is better. Methods that do no use deep networks are listed first, followed by deep clustering algorithms that use fully-connected autoencoders (including the fully-connected configuration of DCC) and deep clustering algorithms that use convolutional autoencoders (including the convolutional configuration of DCC). Results that are within 1% of the highest accuracy achieved by any method are highlighted in bold. DCC performs on par or better than prior deep clustering formulations, without relying on a priori knowledge of the number of ground-truth clusters.

## 5 ANALYSIS

**Importance of joint optimization.** We now analyze the importance of performing dimensionality reduction and clustering jointly, versus performing dimensionality reduction and then clustering the embedded data. To this end, we use the same SDAE architecture and training procedure as fully-connected DCC. We optimize the autoencoder but do not optimize the full DCC objective. This yields a standard nonlinear embedding, using the same autoencoder that is used by DCC, into a space with the same reduced dimensionality $d$. In this space, we apply a number of clustering algorithms: $k$-means++, AC-W, DBSCAN, SEC, LDMGI, GDL, and RCC. The results are shown in Table 2 (top).

These results should be compared to results reported in Table 1. The comparison shows that the accuracy of the baseline algorithms benefits from dimensionality reduction. However, in all cases their accuracy is still lower than that attained by DCC using joint optimization. Furthermore, although RCC and DCC share the same underlying nearest-neighbor graph construction and a similar clustering loss, the performance of DCC far surpasses that achieved by stagewise SDAE embedding followed by RCC. Note also that the relative performance of most baselines drops on Coil100 and YaleB. We hypothesize that the fully-connected SDAE is limited in its ability to discover a good low-dimensional embedding for very high-dimensional image datasets (tens of thousands of dimensions for Coil100 and YaleB).

Next, we show the performance of the same clustering algorithms when they are applied in the reduced space produced by DCC. These results are reported in Table 2 (bottom). In comparison to Table 2 (top), the performance of all algorithms improves significantly and some results are now on par or better than the results of DCC as reported in Table 1. The improvement for $k$-means++, Ward, and DBSCAN is particularly striking. This indicates that the performance of many clustering algorithms can be improved by first optimizing a low-dimensional embedding using DCC and then clustering in the learned embedding space.

| Dataset | $k$-means++ | AC-W | DBSCAN | SEC | LDMGI | GDL | RCC | DCC |
|---|---|---|---|---|---|---|---|---|
| Clustering in a reduced space learned by SDAE | | | | | | | | |
| MNIST | 0.669 | 0.784 | 0.115 | n/a | 0.828 | n/a | 0.881 | 0.912 |
| Coil100 | 0.333 | 0.336 | 0.170 | 0.384 | 0.318 | 0.335 | 0.589 | 0.952 |
| YTF | 0.764 | 0.831 | 0.595 | 0.527 | 0.612 | 0.699 | 0.827 | 0.877 |
| YaleB | 0.673 | 0.688 | 0.503 | 0.493 | 0.676 | 0.742 | 0.812 | 0.955 |
| Reuters | 0.501 | 0.494 | 0.042 | 0.435 | 0.517 | 0.488 | 0.542 | 0.572 |
| RCV1 | 0.454 | 0.430 | 0.075 | 0.442 | 0.060 | 0.055 | 0.410 | 0.495 |
| Clustering in a reduced space learned by DCC | | | | | | | | |
| MNIST | 0.880 | 0.883 | 0.890 | n/a | 0.868 | n/a | 0.912 | 0.912 |
| Coil100 | 0.947 | 0.947 | 0.569 | 0.604 | 0.919 | 0.915 | 0.891 | 0.952 |
| YTF | 0.845 | 0.841 | 0.896 | 0.586 | 0.762 | 0.658 | 0.879 | 0.877 |
| YaleB | 0.811 | 0.809 | 0.809 | 0.584 | 0.815 | 0.660 | 0.814 | 0.955 |
| Reuters | 0.553 | 0.554 | 0.560 | 0.479 | 0.586 | 0.401 | 0.581 | 0.572 |
| RCV1 | 0.536 | 0.472 | 0.496 | 0.452 | 0.178 | 0.326 | 0.474 | 0.495 |

Table 2: Importance of joint optimization. This table shows the accuracy (AMI) achieved by running prior clustering algorithms on a low-dimensional embedding of the data. For reference, DCC results from Table 1 are also listed. **Top:** The embedding is performed using the same autoencoder architecture as used by fully-connected DCC, into the same target space. However, dimensionality reduction and clustering are performed separately. Clustering accuracy is much lower than the accuracy achieved by DCC. **Bottom:** Here clustering is performed in the reduced space discovered by DCC. The performance of all clustering algorithms improves significantly.

**Visualization.** A visualization is provided in Figure 1. Here we used Barnes-Hut t-SNE (van der Maaten & Hinton, 2008; van der Maaten, 2014) to visualize a randomly sampled subset of 10K datapoints from the MNIST dataset. We show the original dataset, the dataset embedded by the SDAE into $\mathbb{R}^d$ (optimized for dimensionality reduction), and the embedding into $\mathbb{R}^d$ produced by DCC. As shown in the figure, the embedding produced by DCC is characterized by well-defined, clearly separated clusters. The clusters strongly correspond to the ground-truth classes (coded by color in the figure), but were discovered with no supervision.

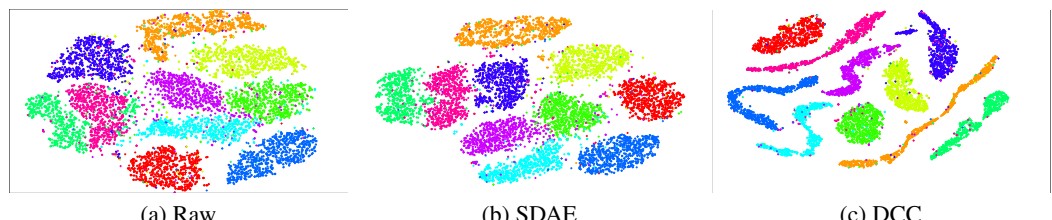

|          (a) Raw          |          (b) SDAE          |          (c) DCC          |

Figure 1: Effect of joint dimensionality reduction and clustering on the embedding. (a) A randomly sampled subset of 10K points from the MNIST dataset, visualized using t-SNE. (b) An embedding of these points into $\mathbb{R}^d$, performed by an SDAE that is optimized for dimensionality reduction. (c) An embedding of the same points by the same network, optimized with the DCC objective. When optimized for joint dimensionality reduction and clustering, the network produces an embedding with clearly separated clusters. Best viewed in color.

**Robustness to dimensionality of the latent space.** Next we study the robustness of DCC to the dimensionality $d$ of the latent space. For this experiment, we consider fully-connected DCC. We vary $d$ between 5 and 60 and measure AMI on the MNIST and Reuters datasets. For comparison, we report the performance of DEC, which uses the same autoencoder architecture, as well as the accuracy attained by running $k$-means++ on the output of the SDAE, optimized for dimensionality reduction. The results are shown in Figure 2.

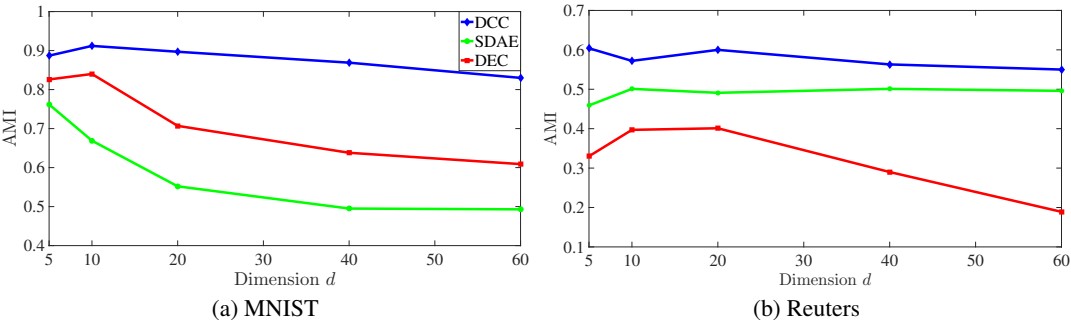

Figure 2: Robustness to dimensionality of the latent space. Clustering accuracy (AMI) as a function of the dimensionality $d$. Best viewed in color.

The results yield two conclusions. First, the accuracy of DCC, DEC, and SDAE+$k$-means gradually decreases as the dimensionality $d$ increases. This supports the common view that clustering becomes progressively harder as the dimensionality of the data increases. Second, the results demonstrate that DCC is more robust to increased dimensionality than DEC and SDAE. For example, on MNIST, as the dimensionality $d$ changes from 5 to 60, the accuracy of DEC and SDAE drops by 28% and 35%, respectively, while the accuracy of DCC decreases by only 9%. When $d = 60$, the accuracy attained by DCC is higher than the accuracy attained by DEC and SDAE by 27% and 40%, respectively.

## 6 CONCLUSION

We have presented a clustering algorithm that combines nonlinear dimensionality reduction and clustering. Dimensionality reduction is performed by a deep network that embeds the data into a lower-dimensional space. The embedding is optimized as part of the clustering process and the resulting network produces clustered data. The presented algorithm does not rely on a priori knowledge of the number of ground-truth clusters. Nonlinear dimensionality reduction and clustering are performed by optimizing a global continuous objective using scalable gradient-based solvers.

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

## APPENDIX

## A DATASETS

**MNIST** (LeCun et al., 1998): This is a popular dataset containing 70,000 images of handwritten digits. Each image is of size $28 \times 28$ (784 dimensions). The data is categorized into 10 classes.

**Coil100** (Nene et al., 1996): This dataset consists of 7,200 images of 100 object categories, each captured from 72 poses. Each RGB image is of size $128 \times 128$ (49,152 dimensions).

**YouTube Faces** (Wolf et al., 2011): The YTF dataset contains videos of faces. We use all the video frames of the first 40 subjects sorted in chronological order. Each frame is an RGB image of size $55 \times 55$. The number of datapoints is 10,056 and the dimensionality is 9,075.

**YaleB** (Georghiades et al., 2001): This dataset contains 2,414 images of faces of 28 human subjects taken under different lightning condition. Each image is of size $192 \times 168$ (32,256 dimensions).

**Reuters**: This is a popular dataset comprising 21,578 Reuters news articles. We consider the Modified Apte split, which yields a total of 9,082 articles. TF-IDF features on the 2,000 most frequently occurring word stems are computed and normalized. The dimensionality of the data is thus 2,000.

**RCV1** (Lewis et al., 2004): This is a document dataset comprising 800,000 Reuters newswire articles. Only the four root categories are considered and all articles labeled with more than one root category are pruned. We report results on a randomly sampled subset of 10,000 articles. 2,000 TF-IDF features were extracted as in the case of the Reuters dataset.

## B    CONVOLUTIONAL NETWORK ARCHITECTURE

Table 3 summarizes the architecture of the convolutional encoder used for the convolutional configuration of DCC. Convolutional kernels are applied with a stride of two. The encoder is followed by a fully-connected layer with output dimension $d$ and a convolutional decoder with kernel size that matches the output dimension of `conv5`. The decoder architecture mirrors the encoder and the output from each layer is appropriately zero-padded to match the input size of the corresponding encoding layer. All convolutional and transposed convolutional layers are followed by batch normalization and rectified linear units (Ioffe & Szegedy, 2015; Nair & Hinton, 2010).

|        | MNIST        | Coil100      | YTF          | YaleB        |
|--------|--------------|--------------|--------------|--------------|
| conv1  | $4 \times 4$ | $4 \times 4$ | $4 \times 4$ | $4 \times 4$ |
| conv2  | $5 \times 5$ | $5 \times 5$ | $5 \times 5$ | $5 \times 5$ |
| conv3  | $5 \times 5$ | $5 \times 5$ | $5 \times 5$ | $5 \times 5$ |
| conv4  | –            | $5 \times 5$ | $5 \times 5$ | $5 \times 5$ |
| conv5  | –            | $5 \times 5$ | –            | $5 \times 5$ |
| output | $4 \times 4$ | $4 \times 4$ | $4 \times 4$ | $6 \times 6$ |

Table 3: Convolutional encoder architecture.

## C    HYPERPARAMETERS

DCC uses three hyperparameters: the nearest neighbor graph (mkNN) parameter $k$, the embedding dimensionality $d$, and the update period $M$ for graduated nonconvexity. For fair comparison to RCC and RCC-DR, we fix $k = 10$ (the setting used in Shah & Koltun (2017)). The other two hyperparameters were set to $d = 10$ and $M = 20$ based on grid search on MNIST. The hyperparameters are fixed at these values across all datasets. No dataset-specific tuning is done. However, note that the hyperparameter $M$ is architecture-specific. We set $M = 10$ for convolutional autoencoders and it is varied for varying dimensionality $d$ during the controlled experiment reported in Figures 2 and 3. The other hyperparameters such as $\lambda, \delta_i, \mu_i$ are set automatically as described in Sections 3.2 and 3.3 and in Shah & Koltun (2017).

## D    ACC MEASURE

For completeness, we report results according to the ACC measure. Table 4 provides the ACC counterpart to Table 1. Figure 3 provides the ACC counterpart to Figure 2.

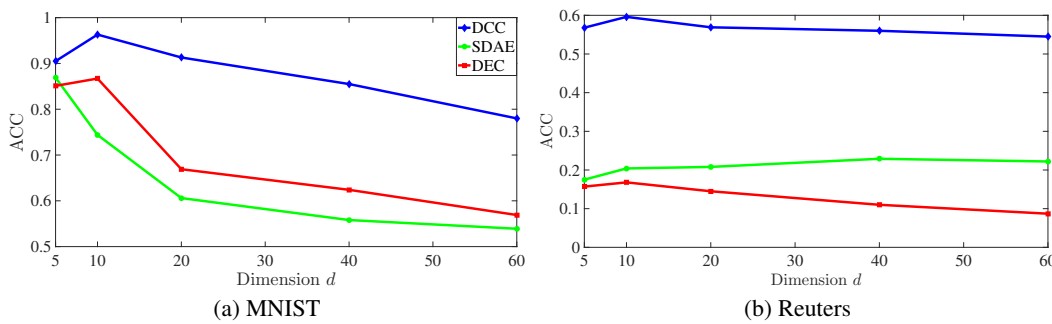

(a) MNIST                      (b) Reuters

Figure 3: Clustering accuracy (ACC) as a function of the dimensionality $d$ of the latent space. This is the ACC counterpart to Figure 2. Best viewed in color.

| Algorithm | MNIST | Coil100 | YTF | YaleB | Reuters | RCV1 |
|---|---|---|---|---|---|---|
| $k$-means++ | 0.532 | 0.621 | 0.624 | 0.514 | 0.236 | 0.529 |
| AC-W | 0.571 | 0.697 | 0.647 | 0.614 | 0.261 | 0.554 |
| DBSCAN | 0.000 | 0.921 | 0.675 | 0.632 | 0.700 | 0.571 |
| SEC | 0.545 | 0.648 | 0.562 | 0.721 | 0.434 | 0.425 |
| LDMGI | 0.723 | 0.763 | 0.332 | 0.901 | 0.465 | 0.667 |
| GDL | n/a | 0.825 | 0.497 | 0.783 | 0.463 | 0.444 |
| RCC | 0.876 | 0.831 | 0.484 | 0.939 | 0.381 | 0.356 |
| RCC-DR | 0.698 | 0.825 | 0.579 | 0.945 | 0.437 | 0.676 |
| Fully Connected | | | | | | |
| DCN | 0.560 | 0.620 | 0.620 | 0.430 | 0.220 | **0.730** |
| DEC | 0.867 | 0.815 | 0.643 | 0.027 | 0.168 | 0.683 |
| DCC | **0.962** | 0.842 | 0.605 | 0.861 | **0.596** | 0.563 |
| Fully Convolutional | | | | | | |
| JULE | 0.800 | **0.911** | 0.342 | **0.970** | – | – |
| DEPICT | **0.968** | 0.420 | 0.586 | **0.965** | – | – |
| DCC | **0.963** | 0.858 | **0.699** | 0.964 | – | – |

Table 4: Clustering accuracy of DCC and 12 baselines, measured by ACC. Higher is better. This is the ACC counterpart to Table 1.

## E  NMI MEASURE

We also report results according to the NMI measure. Table 5 provides the NMI counterpart to Table 1.

| Algorithm | MNIST | Coil100 | YTF | YaleB | Reuters | RCV1 |
|---|---|---|---|---|---|---|
| $k$-means++ | 0.500 | 0.835 | 0.788 | 0.650 | 0.536 | 0.355 |
| AC-W | 0.679 | 0.876 | 0.806 | 0.788 | 0.492 | 0.364 |
| DBSCAN | 0.000 | 0.458 | 0.756 | 0.535 | 0.022 | 0.017 |
| SEC | 0.469 | 0.872 | 0.760 | 0.863 | 0.498 | 0.069 |
| LDMGI | 0.761 | 0.906 | 0.532 | 0.950 | 0.523 | 0.382 |
| GDL | n/a | 0.965 | 0.664 | 0.931 | 0.401 | 0.020 |
| RCC | 0.893 | 0.963 | 0.850 | 0.978 | 0.556 | 0.138 |
| RCC-DR | 0.827 | 0.963 | 0.882 | 0.976 | 0.553 | 0.442 |
| Fully-connected | | | | | | |
| DCN | 0.570 | 0.830 | 0.810 | 0.630 | 0.460 | 0.470 |
| DEC | 0.853 | 0.645 | 0.811 | 0.000 | 0.409 | **0.504** |
| DCC | **0.912** | 0.961 | 0.886 | 0.959 | **0.588** | 0.498 |
| Convolutional | | | | | | |
| JULE | 0.900 | **0.983** | 0.587 | **0.991** | – | – |
| DEPICT | **0.919** | 0.678 | 0.790 | **0.990** | – | – |
| DCC | **0.915** | 0.967 | **0.908** | 0.987 | – | – |

Table 5: Clustering accuracy of DCC and 12 baselines, measured by NMI. Higher is better. This is the NMI counterpart to Table 1.

