# OpenReview forum: "Deep Continuous Clustering"
_ICLR.cc/2018/Conference — Reject_

### Official Review · AnonReviewer1 · 2017-11-21
**Authors of this paper presented a clustering algorithm by jointly solving deep autoencoder and clustering as a global continuous objective. Experiments demonstrate better results than state-of-the-art clustering schemas.**

**Rating:** 6
**Confidence:** 3

**Review:**

As authors stated, the proposed DCC is very similar to RCC-DR (Shah & Koltun, 2007). The only difference in (3) from RCC-DR is the decoding part, which is replaced by autoencoder instead of linear transformation used in RCC-DR. Authors claimed that there are three major differences. However, due to the highly nonconvex properties of both formulations, the last two differences hardly support the advantages of the proposed DCC comparing with RCC-DR because the solutions obtained by both optimization approaches are local solutions, unless authors can claim that the gradient-based solver is better than alternating approach in RCC-DR. Hence, DCC is just a simple extension of RCC-DR.

In Section 3.2, how does the optimization algorithm handle the equality constraints in (5)? It is unclear why the existing autoencoder solver can be used to solve (3) or (5). It seems that the first term in (5) corresponds to the objective of autoencoder, but the last two terms added lead to different objective with respect to variables y. It is better to clarify the correctness of the optimization algorithm.

Authors claimed that the proposed method avoid discrete reconfiguration of the objective that characterize prior clustering algorithms, and it does not rely on a priori knowledge of the number of ground-truth clusters. However, it seems not true since the graph construction at every epoch depends on the initial parameter delta_2 and the graph is constructed such that f_{i,j}=1 if distance is less than delta_2. As a result, delta_2 is a fixed threshold for graph construction, so it is indirectly related to the number of clusters generated. In the experiments, authors set it as the mean of the bottom 1% of the pairwise distances in E at initialization, and clustering assignment is given by connected component in the last graph. This parameter might be sensitive to the final results.

Many terms in the paper are not well explained. For example, in (1), theta are treated as parameters to optimize, but what is the theta used for? Does the Omega related to encoder and decoder of the parameters in autoencoder. What is the scaled Geman-McClure function? Any reference? Why should this estimator be used?

From the visualization results in Figure 1, it is interesting to see that K-means++ can achieve much better results on the space learned by DCC than that by SDAE from Table 2. In Figure 1, the embedding by SDAE (Figure 1(b)) seems more suitable for kmeans-like algorithm than DCC (Figure 1(c)). That is the reason why connected component is used for cluster assignment in DCC, not kmeans. The results between Table 2 and Figure 1 might be interesting to investigate.

---

> ### Author Response · Authors · 2017-12-15
> **Response to review**
>
> Thank you for your work on the paper. We respond to each comment below.
>
> Q: As authors stated, the proposed DCC is very similar to RCC-DR (Shah & Koltun, 2007). The only difference in (3) from RCC-DR is the decoding part, which is replaced by autoencoder instead of linear transformation used in RCC-DR. Authors claimed that there are three major differences. However, due to the highly nonconvex properties of both formulations, the last two differences hardly support the advantages of the proposed DCC comparing with RCC-DR because the solutions obtained by both optimization approaches are local solutions, unless authors can claim that the gradient-based solver is better than alternating approach in RCC-DR. Hence, DCC is just a simple extension of RCC-DR.
>
> A: We do see all three advantages as valuable. (Nonlinear embedding, direct optimization of the joint objective, and scalable optimization that does not rely on least-squares.) Aside from the more expressive nonlinear embedding, the key advantage is that DCC simultaneously optimizes the global objective over all variables, while RCC-DR is an alternating EM-like algorithm.
>
>
> Q: In Section 3.2, how does the optimization algorithm handle the equality constraints in (5)? It is unclear why the existing autoencoder solver can be used to solve (3) or (5). It seems that the first term in (5) corresponds to the objective of autoencoder, but the last two terms added lead to different objective with respect to variables y. It is better to clarify the correctness of the optimization algorithm.
>
> A: The equality constraints in (1), (3), and (5) are written out as constraints only for the sake of exposition. In fact these are not distinct constraints: Instead of Y, we simply use F_\Theta(X) inside the relevant terms. Y is only used for exposition.
>
>
> Q: Authors claimed that the proposed method avoid discrete reconfiguration of the objective that characterize prior clustering algorithms, and it does not rely on a priori knowledge of the number of ground-truth clusters. However, it seems not true since the graph construction at every epoch depends on the initial parameter delta_2 and the graph is constructed such that f_{i,j}=1 if distance is less than delta_2. As a result, delta_2 is a fixed threshold for graph construction, so it is indirectly related to the number of clusters generated. In the experiments, authors set it as the mean of the bottom 1% of the pairwise distances in E at initialization, and clustering assignment is given by connected component in the last graph. This parameter might be sensitive to the final results.
>
> A: By discrete reconfigurations of the objective we meant that the objective is influenced by the intermediate cluster assignments. In DCC, the graph G is only used to evaluate the stopping criterion (“Should the optimization stop now?”). It does not affect the objective itself. The graph G does not influence or modify the objective function in any way. So there is no discrete reconfiguration of the objective.
>
>
> Q: Many terms in the paper are not well explained. For example, in (1), theta are treated as parameters to optimize, but what is the theta used for? Does the Omega related to encoder and decoder of the parameters in autoencoder. What is the scaled Geman-McClure function? Any reference? Why should this estimator be used?
>
> A: \theta and \omega are the encoder and decoder network weights, respectively. \Omega is simply the notation used for representing the union of parameters in both networks. The revision we posted includes the definition of the scaled Geman-McClure penalty, which is adopted from RCC.
>
>
> Q: From the visualization results in Figure 1, it is interesting to see that K-means++ can achieve much better results on the space learned by DCC than that by SDAE from Table 2. In Figure 1, the embedding by SDAE (Figure 1(b)) seems more suitable for kmeans-like algorithm than DCC (Figure 1(c)). That is the reason why connected component is used for cluster assignment in DCC, not kmeans. The results between Table 2 and Figure 1 might be interesting to investigate.
>
> A: That is an interesting suggestion. It is possible that k-means or a similar algorithm are well-suited for SDAE output. That being said, we caution against drawing major conclusions from a two-dimensional embedding of high-dimensional data. We avoid using k-means because it requires knowing the number of clusters a priory. Not requiring such knowledge is a major advantage of the RCC/DCC family of algorithms.

---

### Official Review · AnonReviewer3 · 2017-11-23
**A continuous relaxation for clustering with deep autoencoder**

**Rating:** 3
**Confidence:** 5

**Review:**

This paper presents a clustering method in latent space. The work extends a previous approach (Shah & Koltun 2017) which employs a continuous relaxation of the clustering assignments. The proposed method is tested on several image and text data sets.

However, the work has a number of problems and unclear points.

1) There is no theoretical guarantee that RCC or DCC can give good clusterings. The second term in Eq. 2 will pull z's closer but it can also wrongly place data points from different clusters nearby.

2) The method uses an autoencoder with elementwise least square loss. This is not suitable for data sets such as images and time series.

3) Please elaborate "redesending M-estimator" in Section 2. Also, please explicitly write out what are rho_1 and rho_2 in the experiments.

4) The method requires many extra hyperparameters lambda, delta_1, delta_2. Users have to set them by ad hoc heuristics.

5) In each epoch, the method has to construct the graph G (the last paragraph in Page 4) over all z pairs.  This is expensive. The author didn't give any running time estimation in theory or in experiments.

6) The experimental results are not convincing. For MNIST its best accuracy is only 0.912. Existing methods for this data set have achieve 0.97 accuracy. See for example [Ref1,Ref2,Ref3]. For RCV1, [Ref2] gives 0.54, but here it is only 0.495.

7) Figure 1 gives a weird result. There is no known evidence that MNIST clusters intrinsically distribute like snakes. They must be some wrong artefacts introduced by the proposed method. Actually t-SNE with MNIST pixels is not bad at all. See [Ref4].

8) It is unknown how to set the number of clusters in proposed method.


[Ref1] Zhirong Yang, Tele Hao, Onur Dikmen, Xi Chen, Erkki Oja. Clustering by Nonnegative Matrix Factorization Using Graph Random Walk. In NIPS 2012.
[Ref2] Xavier Bresson, Thomas Laurent, David Uminsky, James von Brecht. Multiclass Total Variation Clustering. In NIPS 2013.
[Ref3] Zhirong Yang, Jukka Corander and Erkki Oja. Low-Rank Doubly Stochastic Matrix Decomposition for Cluster Analysis. Journal of Machine Learning Research, 17(187): 1-25, 2016.
[Ref4] https://sites.google.com/site/neighborembedding/mnist

Confidence: 5: The reviewer is absolutely certain that the evaluation is correct and very familiar with the relevant literature

---

> ### Author Response · Authors · 2017-12-15
> **Response to review**
>
> Q: 1) There is no theoretical guarantee that RCC or DCC can give good clusterings. The second term in Eq. 2 will pull z's closer but it can also wrongly place data points from different clusters nearby.
>
> A: Clustering is NP-hard. No published deep clustering algorithm provides theoretical guarantees. Constructions exist that will make both classic and deep clustering algorithms fail. For example, k-means can get stuck in a local minimum that has an arbitrarily bad cost. See, for example, Sanjoy Dasgupta, “The hardness of k-means clustering”, 2008. Due to the intractability of NP-hard problems, clustering algorithms are evaluated in terms of empirical performance on standard datasets.
>
>
> Q: 2) The method uses an autoencoder with elementwise least square loss. This is not suitable for data sets such as images and time series.
>
> A: The reviewer is mistaken. Autoencoders are commonly applied to images. Our experiments include multiple datasets of images.
>
>
> Q: 3) Please elaborate "redesending M-estimator" in Section 2. Also, please explicitly write out what are rho_1 and rho_2 in the experiments.
>
> A: We explicitly define rho_1 and rho_2 in the revision. “Redescending” is a standard term in robust statistics. See (Shah & Koltun, 2017) and a substantial body of statistics literature.
>
>
> Q: 4) The method requires many extra hyperparameters lambda, delta_1, delta_2. Users have to set them by ad hoc heuristics.
>
> A: The reviewer is mistaken. None of these hyperparameters (lambda, delta_1, delta_2) have to be set “by ad hoc heuristics”. They are set automatically using principled formulae. These formulae are given in (Shah & Koltun, 2017).
>
>
> Q: 5) In each epoch, the method has to construct the graph G (the last paragraph in Page 4) over all z pairs.  This is expensive. The author didn't give any running time estimation in theory or in experiments.
>
> A: The reviewer is mistaken. As stated in the paper, the graph G is only constructed “once the continuation scheme is completed”, once per epoch, to evaluate the stopping criterion. The graph is constructed only over z-pairs that are already in the graph E. And this construction is not expensive at all. For example, on MNIST it takes ~1.1 sec using the scipy package. (Note also that this step is also part of the RCC algorithm.)
>
> In terms of runtime, we do not claim any major advantage, but the runtime is not bad. For instance, on MNIST (the largest dataset considered), the total runtime of conv-DCC is 9030 sec. For DEPICT, this runtime is 12072 sec and for JULE it is 172058 sec. The runtime of DCC is mildly better than DEPICT and more than an order of magnitude better than JULE.
>
>
> Q: 6) The experimental results are not convincing. For MNIST its best accuracy is only 0.912. Existing methods for this data set have achieve 0.97 accuracy. See for example [Ref1,Ref2,Ref3]. For RCV1, [Ref2] gives 0.54, but here it is only 0.495.
>
> A: The reviewer is mistaken. The numbers reported in our paper are according to the AMI metric. The 0.97 accuracy in [Ref1] is using the `purity’ metric. These metrics are substantially different and are not comparable. By way of background, note that the purity metric is biased towards finer-grained clusterings. For example, if each datapoint is set to be a cluster in itself, then the purity of the clustering is 1.0. Purity is a bad metric that is easy to game. It is avoided in recent serious work on clustering.
>
>
> Q: 7) Figure 1 gives a weird result. There is no known evidence that MNIST clusters intrinsically distribute like snakes. They must be some wrong artefacts introduced by the proposed method. Actually t-SNE with MNIST pixels is not bad at all. See [Ref4].
>
> A: First, the t-SNE figure in [Ref4] is plotted using weighted t-SNE whereas we use bh-t-SNE. Second, note that the elongated (“snake-like”) structure also appears in the embedding output of other clustering algorithm (see, e.g., Figure 4 in (Shah & Koltun, 2017)). Third, one should not read much into the detailed planar shapes formed by embedding high-dimensional pointsets into the plane. The clean separation of the clusters is more relevant than the detailed shapes they form in the planar embedding.
>
>
> Q: 8) It is unknown how to set the number of clusters in proposed method.
>
> A: DCC does not require setting the number of clusters in advance. As explained in the paper, this is one of the key advantages of the presented algorithm compared to prior deep clustering algorithms.

---

> > ### Comment · AnonReviewer3 · 2018-01-11
> > **The answers do not clarify the issues. Severe drawbacks remain in the work.**
> >
> > The response is disappointing. Keep saying that I am mistaken will not clarify the issues.
> > 1) The authors admit there is no theoretical guarantee. I am not asking about hardness. So it has nothing to do with NP-hard. If the work indeed has breakthrough, it should contain some theoretical guarantee or at least explaination that the method must lead to wide margins between the clusters. Unfortunately the authors simply avoid answering this.
> > 2) It is well known that pixelwise distance is sensitive for comparing two images. Therefore it is also a known drawback in VAE. The current work inherits the same drawback.
> > 3) I don't think "redescending M-estimator" is well-known in ICLR. Elaborating the term "redesending M-estimator" can help readers understand the method.
> > 4) The hyperpameters are calculated in a manner without theoretical guarantee or explanation. How can you say that these are "principled". I don't find any grounds that these calculation corresponds to their optimal choice.
> > 5) The running time analysis should be added to the paper. Now it is completely missing. According to the rebuttal, the proposed method is significantly slower than those in [Ref1-3].
> > 6) The answer is simply wrong. Fixing the number of clusters, purity can measure clustering accuracy. Therefore the proposed method reads inferior in accuarcy. Moreover, DCD in [Ref3] does not favor more clusters. It can automatically choose the number of clusters.
> > 7) bh-t-SNE will not give snake-like visualization of MNIST. There must be something wrong in the presented results.
> > 8) There is no evidence in the paper that the proposed method can give the right number of clusters. Moreover, the resulting number of clusters depends on the value of delta_2, which is tricky to set.

---

> > > ### Author Response · Authors · 2018-01-13
> > > **response to reviewer's post**
> > >
> > > We will not try to change this reviewer's mind. If the ACs wish, we will post a detailed point-by-point rebuttal to the reviewer's latest post.
> > >
> > > As an illustrative example, we briefly again address point (6), because the reviewer's latest post is so strongly worded on this point. (The reviewer's commentary begins with "The answer is simply wrong.")
> > >
> > > First, note that the reviewer appears to have shifted from equating "purity" with "AMI" (in the initial review) to equating "purity" with "ACC" (in the latest comment). The reviewer's original comment clearly points to our AMI numbers and states that they are weaker than "purity" numbers reported in other papers. But these are completely different metrics, the numbers are incomparable. (And the use of the two incomparable metrics is abundantly clear in both papers, so the reviewer's error is rather glaring.)
> > >
> > > In the reviewer's latest post, the reviewer does not acknowledge the mistake but rather shifts to discussing "clustering accuracy". If we interpret the reviewer's comment correctly, the reviewer is now alluding to ACC, a different clustering measure which is reported for completeness in our supplement. Yet here again the reviewer is mistaken, because the "purity" measure does not reduce to ACC even when the number of clusters is fixed. No known formula exists for converting AMI or ACC to purity. As far as we can tell, the reviewer's conclusion -- "Therefore the proposed method reads inferior in accuarcy [sic]" -- is baseless. If the ACs wish, we can break down the definitions of AMI, ACC, and Purity in detail, and show further, based on the formulas, that the reviewer's statements are unfounded.

---

> > > > ### Comment · AnonReviewer3 · 2018-01-13
> > > > **Some more comments**
> > > >
> > > > No, argument about canonical ACC definition is not my focus. Now it reads that the advantages claimed in the paper is conditional on the choice of clustering performance measure. Even if AMI is used, it is still hard to convince that the proposed method brings significant improvement because the authors refuse to compare to recent clustering approaches.
> > > >
> > > > What I said "simply wrong" means that the authors' rebuttal about various number of clusters is incorrect. I didn't intend to distinguish this point. Actually, lack of convincing evidence about significant performance improvement, i.e. Point 6 in my review, is only one of listed weak points of the work.
> > > >
> > > > Still about the t-SNE visualization of MNIST. If a digit cluster distributes like a snake, it means that the variation of the images of the digit is intrinsically one-dimensional. This is counter-intuitive. I doubt the t-SNE algorithm is not converged yet.
> > > >
> > > > d=10 is another tricky setting. We don't know whether this is suitable in general.

---

### Official Review · AnonReviewer2 · 2017-11-25
**The authors proposed a new clustering algorithm named deep continuous clustering (DCC) that integrates autoencoder into continuous clustering. The paper is interesting and could be improved by increasing the usability of the method.**

**Rating:** 7
**Confidence:** 4

**Review:**

The authors proposed a new clustering algorithm named deep continuous clustering (DCC) that integrates autoencoder into continuous clustering. As a variant of  continuous clustering (RCC), DCC formed a global continuous objective for joint nonlinear dimensionality reduction and clustering. The objective can be directly optimized using SGD like method. Extensive experiments on image and document datasets show the effectiveness of DCC. However, part of experiments are not comprehensive enough.

The idea of integrating autoencoder with continuous clustering is novel, and the optimization part is quite different. The trick used in the paper (sampling edges but not samples) looks interesting and seems to be effective.

In the following, there are some detailed comments:
1. The paper is well written and easy to follow, except the definition of Geman-McClure function is missing. It is difficult to follow Eq. (6) and (7).
2. Compare DCC to RCC, the pros and cons are obvious. DCC does improve the performance of clustering with the cost of losing robustness. DCC is more sensitive to the hyper-parameters, especially embedding dimensionality d. With a wrong d DCC performs worse than RCC on MNIST and similar on Reuters. Since clustering is one unsupervised learning task. The author should consider heuristics to determine the hyper-parameters. This will increase the usability of the proposed method.
3. However, the comparison to the DL based partners are not comprehensive enough, especially JULE and DEPICT on image clustering. Firstly, the authors only reported AMI and ACC, but not NMI that is reported in JULE. For a fair comparison, NMI results should be included. Secondly, the reported results do not agree with the one in original publication. For example, JULE reported ACC of 0.964 and 0.684 on MNIST and YTF. However, in the appendix the numbers are 0.800 and 0.342 respectively. Compared to the reported number in JULE paper, DCC is not significantly better.

In general, the paper is interesting and proposed method seems to be promising. I would vote for accept if my concerns can be addressed.

The author's respond address part of my concerns, so I have adjusted my rating.

---

> ### Author Response · Authors · 2017-12-15
> **Response to review**
>
> Thank you for your work on the paper. We respond to each comment below.
>
> Q: 1. The paper is well written and easy to follow, except the definition of Geman-McClure function is missing. It is difficult to follow Eq. (6) and (7).
>
> A: Thanks for pointing this out. We addressed this in the revision.
>
>
> Q: 2. Compare DCC to RCC, the pros and cons are obvious. DCC does improve the performance of clustering with the cost of losing robustness. DCC is more sensitive to the hyper-parameters, especially embedding dimensionality d. With a wrong d DCC performs worse than RCC on MNIST and similar on Reuters. Since clustering is one unsupervised learning task. The author should consider heuristics to determine the hyper-parameters. This will increase the usability of the proposed method.
>
> A: We have clarified hyperparameter settings in the revision. In brief, DCC uses three hyperparameters: the nearest neighbor graph parameter ‘k’, the embedding dimensionality ‘d’, and the graduated nonconvexity parameter ‘M’. For fair comparison to RCC and RCC-DR, we fix k=10 (the setting used in (Shah & Koltun, 2017)). The other two hyperparameters were set to d=10 and M=20 based on grid search on MNIST. The hyperparameters are fixed at these values across all datasets. No dataset-specific tuning is done. Other hyperparameters, such as \lambda, \delta_i, and \mu_i, are inherited from RCC and are set automatically as described in the RCC paper.
>
>
> Q: 3. However, the comparison to the DL based partners are not comprehensive enough, especially JULE and DEPICT on image clustering. Firstly, the authors only reported AMI and ACC, but not NMI that is reported in JULE. For a fair comparison, NMI results should be included.
>
> A: We have included NMI results in the revision. (Appendix E.)
>
>
> Q: Secondly, the reported results do not agree with the one in original publication. For example, JULE reported ACC of 0.964 and 0.684 on MNIST and YTF. However, in the appendix the numbers are 0.800 and 0.342 respectively. Compared to the reported number in JULE paper, DCC is not significantly better.
>
> A: In order to report AMI measurements, we reran JULE using publicly shared code from the authors. We were unable to reproduce the results on MNIST despite using the preprocessed MNIST data shared by the authors and keeping all other parameters fixed as suggested on the JULE GitHub repo. The JULE article reports NMI on two versions of the algorithm, JULE-SF and JULE-RC. We report numbers for JULE-RC as authors state that this is the slightly better algorithm. In our experiments, the NMI on MNIST for JULE-SF is 0.912 and the NMI for JULE-RC is 0.900. The measured NMI for each dataset is:
>
> 		MNIST	Coil-100 	YTF		YaleB
> JULE-SF	0.912		0.969		0.754		0.994
> JULE-RC	0.900		0.983		0.587		0.991
>
> In running JULE on the YTF dataset, we followed a similar protocol to the RCC paper. This processes the YTF data in a slightly different fashion than in the JULE and DEPICT papers, but we adopt the RCC data preparation protocol for consistency with other baselines and experiments. This data preparation protocol yields 10056 samples from 40 subjects, while the version in the JULE paper 10000 samples from 41 subjects. However, it is hard to believe that this small difference would lead to large changes in the resulting accuracy. For reference, our results on YTF for DEPICT are very close to the results reported in the original DEPICT publication. Finally, please note that DCC achieves similar or better accuracy without any knowledge of the number of clusters, whereas JULE and DEPICT do use a priori knowledge of the ground-truth number of clusters.

---

### Author Response · Authors · 2017-12-15
**Revision and responses to reviews**

We have uploaded a revision that addresses comments brought up in the reviews. In addition, we have posted responses to each individual review. These responses, which address each comment in detail, can be found below.

---

### Author Response · Authors · 2017-12-28
**Any questions or concerns?**

Dear ACs and reviewers,

Do you have any questions? Are there any remaining concerns?

We strongly believe that the work is solid, as demonstrated by the extensive experiments. We would be happy to address any remaining questions or concerns.

Best regards,
The authors

---

> ### Comment · AnonReviewer2 · 2018-01-05
> **Compare to Autoencoder + RCC**
>
> It would be interest to see the comparison to another simple two step baseline, Autoencoder followed by RCC.

---

> > ### Author Response · Authors · 2018-01-06
> > **Re: Compare to Autoencoder + RCC**
> >
> > Good question. In fact this is already in the paper. This comparison is provided in Table 2 (page 8). The top half of this table ("Clustering in a reduced space learned by SDAE") shows the accuracy achieved by running various clustering algorithms, including RCC, in a space learned by an Autoencoder. (For reference, DCC results are also listed, in the last column.) Specifically, compare the second-to-last column (Autoencoder + RCC) to the last column (DCC). The DCC results are much better than Autoencoder + RCC.

---

### Decision · Program_Chairs · 2018-01-29
**ICLR 2018 Conference Acceptance Decision**

**Decision:**

Reject

**Comment:**

After careful consideration, I think that this paper in its current form is just under the threshold for acceptance. Please note that I did take into account the comments, including the reviews and rebuttals, noting where arguments may be inconsistent or misleading.

The paper is a promising extension of RCC, albeit too incremental. Some suggestions that may help for the future:

1) Address the sensitivity remark of reviewer 2. If the hyperparameters were tuned on RCV1 instead of MNIST, would the results across the other datasets remain consistent?

2) Train RCC or RCC-DR in an end-to-end way to gauge the improvement of joint optimization over alternating, as this is one of the novel contributions.

3) Discuss how to automatically tune \lambda and \delta_1 and \delta_2. These may appear in the RCC paper, but it's unclear if the same derivations hold when going to the non-linear case (they may in fact transfer gracefully, it's just not obvious). It would also be helpful for researchers building on DCC.